# Unilateral Palpebral Edema as a Central Sign of Acute Enterobacter-Associated Rhinosinusitis in a 5-Year-Old: A Rare Pediatric Case

**DOI:** 10.3390/reports8020066

**Published:** 2025-05-14

**Authors:** Andrei Osman, Irina Enache, Alice Elena Ghenea, Alexandra Bucătaru, Sidonia Cătălina Vrabie, Ovidiu Mircea Zlatian

**Affiliations:** 1Otorhinolaringology Department, Emergency County Hospital of Craiova, 200642 Craiova, Romania; andrei.osman@umfcv.ro; 2Department of Anatomy and Embriology, University of Medicine and Pharmacy of Craiova, 200349 Craiova, Romania; 3Clinical Laboratory, Emergency County Hospital of Craiova, 200642 Craiova, Romania; alice.ghenea@umfcv.ro (A.E.G.); ovidiu.zlatian@umfcv.ro (O.M.Z.); 4Department of Microbiology, University of Medicine and Pharmacy of Craiova, 200349 Craiova, Romania; 5Doctoral School, University of Medicine and Pharmacy of Craiova, 200349 Craiova, Romania; alexandra.catana95@gmail.com; 6Department of Obstetrics and Gynecology, University of Medicine and Pharmacy of Craiova, 200349 Craiova, Romania

**Keywords:** acute rhinosinusitis, pediatric rhinosinusitis, computed imaging, painful palpebral edema, *Enterobacter* species, endoscopic ethmoidectomy

## Abstract

**Background and Clinical Significance**: Acute pediatric rhinosinusitis is most commonly caused by *Streptococcus pneumoniae*, *Haemophilus influenzae*, and *Moraxella catarrhalis*. The involvement of *Enterobacter* species is rare and typically linked to chronic or nosocomial infections. Typical cases of acute rhinosinusitis in children present with abundant nasal discharge, headache, and fever and are generally managed with systemic antibiotics, nonsteroidal anti-inflammatory drugs (NSAIDs), mucolytics, and topical intranasal treatment. Atypical presentations prompt heightened clinical attention, and depending on the symptoms and patient status, surgical interventions might be considered. **Case Presentation**: We report the case of a previously healthy 5-year-old boy presenting with painful unilateral palpebral edema, minimal ipsilateral nasal discharge, and persistent headache despite standard rhinosinusitis therapy. Imaging tests revealed complete right maxillary sinus opacification. As the clinical response to ceftriaxone and dexamethasone was minimal, we opted for endoscopic sinus surgery. A nasal swab culture identified *Enterobacter* spp. in the nasal discharge. **Conclusions**: Unusual pathogens like *Enterobacter* spp. can cause acute sinusitis in children without prior risk factors. Early surgical intervention and culture-adjusted antimicrobial therapy remain critical for favorable outcomes.

## 1. Introduction and Clinical Significance

The predominant bacterial pathogens in acute sinusitis are typically *Streptococcus pneumoniae*, *Haemophilus influenzae*, and *Moraxella catarrhalis* [1,2]. However, *Enterobacteriaceae*, which includes *Enterobacter* spp., can be identified in some cases of acute rhinosinusitis, particularly in mixed infections or in specific clinical settings [3]. *Enterobacter* spp., as part of the *Enterobacteriaceae* family, are not commonly identified as primary agents of acute sinusitis. The presence of *Enterobacter* in sinusitis cases is more frequently associated with chronic conditions or acute exacerbations of chronic sinusitis rather than primary acute presentations [4,5].

*Enterobacteriaceae* are not reported to be the leading microbial cause of acute sinusitis in adults or children. The most common bacteria isolated in acute cases of sinusitis, as most literature reports specify, are *Streptococcus pneumoniae* and *Haemophilus influenzae* [6]. *Enterobacteriaceae* are more commonly found in chronic sinusitis and acute exacerbations of chronic sinusitis (AECS). In a study of AECS, *Enterobacteriaceae* were among the predominant aerobic bacteria isolated, although they were not the leading bacterial culprits for the cases in this study [7]. The presence of multidrug-resistant *Enterobacteriaceae* in sinusitis cases, particularly chronic ones, complicates treatment and may lead to their increased detection in cultures when standard treatments for rhinosinusitis fail [8]. *Enterobacter* spp. may be part of mixed bacterial infections, which are more common in chronic sinusitis and AECS. These mixed infections often include both aerobic and anaerobic bacteria [7,9].

The presence of *Enterobacter*, especially in chronic sinusitis, poses challenges due to its antibiotic resistance. Empiric therapy seldom succeeds, necessitating culture and sensitivity testing to allow for appropriate antibiotic selection [8]. However, the presence of *Enterobacter* in a new, acute, complicated case of rhinosinusitis will lead to a high degree of clinical suspicion and attention from the otorhinolaryngologist toward his patient, especially if this patient is a pediatric case. *Enterobacter* spp., particularly those producing extended-spectrum beta-lactamase (ESBL), pose significant clinical challenges in pediatric acute sinusitis due to their multidrug-resistant nature, complicating treatment options [10]. Infections with ESBL-producing *Enterobacteriaceae* are associated with higher mortality rates and an increased risk of complications, such as bacteremia, complicated respiratory infections [11], and orbital implications [12].

We report the case of a previously healthy 5-year-old boy with no prior history of rhinosinusitis, chronic illness, or hospitalizations, who presented at the Emergency Department of the Emergency County Hospital in Craiova in November 2024 with unilateral painful palpebral edema, severe frontal and maxillary headache, and minimal rhinorrhea. These symptoms had developed despite 72 h of at-home management with NSAIDs, oral amoxicillin, topical intranasal decongestants, and saline irrigation.

## 2. Case Presentation

We present the case of a 5-year-old boy with no prior history of acute or chronic rhinosinusitis, recurrent respiratory tract infections, immunodeficiency, or previous hospitalizations. The child was brought to the emergency department of the Emergency County Hospital in Craiova in November 2024 by his parents after three days of persistent symptoms that were unresponsive to at-home treatment. The child complained of nasal obstruction, intermittent nasal discharge with a fetid odor, and some facial pain. Initial therapy at home included oral NSAIDs (ibuprofen) and amoxicillin, saline nasal irrigations, and over-the-counter topical nasal decongestants. Despite this course of treatment prescribed by the primary care doctor, the patient had developed progressively worsening unilateral painful palpebral edema in the last 24 h, including progressive headache and dense nasal discharge. These symptoms were accompanied by episodes of diplopia, which the child described while attempting to make out letters or view images on his mother’s smartphone at close range, raising clinical concern for acute rhinosinusitis with potential orbital involvement.

On clinical examination, the child presented with low-grade fever and irritability, with right superior and inferior palpebral edema, tenderness upon digital pressure over the maxillary sinus region, and minimal viscous mucopurulent discharge visible in the right nasal cavity originating from the middle nasal meatus. Nasal endoscopy confirmed the significant congestion and obstruction of the concerned middle meatus and the presence of discharge. Laboratory evaluation revealed leukocytosis with neutrophilia with a total white blood cell count of 19,800/mm^3^, elevated C-reactive protein levels (142 mg/L), and an erythrocyte sedimentation rate of 72 mm/h, all pointing to an acute bacterial infection. Given the atypical presentation, particularly in the absence of abundant nasal discharge, imaging was warranted to confirm the suspicion of sinusitis and assess for early orbital complications. A cranial computed tomography (CT) was performed, which showed complete opacification of the right maxillary sinus, with some mucosal thickening slightly extending into the adjacent ethmoid air cells but no evidence of abscess formation or orbital involvement (Figure 1).

Given the clinical severity of this case and lack of response to pre-existing oral therapy, the patient was admitted to the Pediatric Otorhinolaryngology Department of our hospital. Informed written consent was obtained from the child’s mother. A nasal swab from the middle nasal meatus was obtained before initiating any further antimicrobial therapy, and this was sent for microbiological analysis, including bacterial culture and antimicrobial sensitivity testing. Empiric therapy was initiated with intravenous ceftriaxone (50 mg/kg/day) as a broad-spectrum agent. Additionally, intravenous dexamethasone was administered to address inflammation and prevent the appearance of further orbital complications, while the administration of topical nasal decongestants and the mechanical aspiration of nasal secretions continued.

After 24 h of medical treatment, there was minimal clinical improvement, which was limited to the disappearance of the fever. Local palpebral edema, headache intensity, and nasal discharge remained unchanged (Figure 2). An informed and parent-approved decision was made to proceed with endoscopic sinus surgery under general anesthesia. The surgical approach included a limited anterior ethmoidectomy and maxillary antrostomy via the middle meatus. Intraoperatively, the maxillary sinus cavity was found to be filled with dense, thick, purulent secretions, and the mucosa was hypertrophic and bled easily upon contact with surgical instruments (Figure 2). The purulent material was thoroughly suctioned and evacuated, and the sinus cavity was irrigated with warm saline to ensure the clearance of infectious debris. Hypertrophic tissue was excised and sent for histopathology, although no neoplastic or granulomatous changes were noted.

Postoperatively, silicone-based packing sheets were left in place to maintain middle-meatus patency for 24 h. Ceftriaxone administration was continued, and gentamicin (40 mg every 12 h) was added to the antimicrobial treatment plan for three days due to the suspected risk of Gram-negative organisms. Renal function was closely monitored during dual therapy, with urea and creatinine levels remaining within normal limits throughout hospitalization.

Microbiological culture results became available on the third day of hospitalization and identified *Enterobacter cloacae* as the predominant pathogen in the nasal swab. The isolated strain was analyzed using the automated microbial identification and antimicrobial resistance testing system Vitek2 (Biomerieux, Salt Lake City, UT, USA), which showed susceptibility to ceftriaxone and sensitivity to gentamicin, validating the initiated combined therapy, along with susceptibility to penicillin, and second and third class cephalosporins and quinolones. Although ESBL production was not phenotypically confirmed, the presence of inducible resistance genes, including markers such as *ctxM15*, was detected through the genetic analysis performed on the multiplex polymerase chain reaction (PCR) system Unyvero (Curetis GmbH, Holzgerlingen, Germany). These findings highlighted the potential for antimicrobial resistance development and warranted careful post-discharge monitoring and follow-up.

The patient exhibited progressive clinical improvement within 24 h following endoscopic sinus surgery, with near-complete resolution of periorbital edema and the full remission of headache. During hospitalization, our team performed daily postoperative nasal care using gentle suctioning to clear residual secretions and maintain a patent and clean surgical field. This approach, in conjunction with ongoing systemic antibiotic therapy, contributed to rapid recovery. The patient was discharged on the seventh postoperative day with normalized inflammatory markers and was prescribed at-home saline nasal irrigation and mucolytic therapy. Upon discharge, particular care was placed on educating both the patient and the parents regarding proper hand hygiene and bathroom sanitation practices. Given the suspected community-acquired nature of the *Enterobacter* infection, we suspected likely person-to-person transmission via inadequate hygiene. Preventive counseling was considered an essential component of post-hospitalization care. Endoscopic follow-up at 15 and 30 days post-surgery demonstrated complete mucosal healing of the sinonasal cavity without evidence of ongoing inflammation or purulent discharge. Owing to the lack of symptoms and a clean endoscopic field, the control CT scan was deferred. To lower the risk of dysbiosis, the patient was started on an intranasal probiotic spray containing *Streptococcus salivarius* following the 30-day period of saline nasal irrigations. The recommended plan consisted of one spray per nostril, administered twice daily for seven consecutive days, followed by a 21-day pause, and repeated in cycles over a total period of three months. After completing the regimen, a follow-up nasal endoscopy revealed well-healed mucosa with no signs of residual inflammation or discharge. Additionally, the final nasal swab culture obtained was negative for pathogenic bacteria.

## 3. Discussion

The genus *Enterobacter* includes Gram-negative, rod-shaped bacteria within the *Enterobacteriaceae* family. These organisms are facultatively anaerobic, non-spore-forming, and motile via peritrichous flagella. They are urease-positive and capable of fermenting lactose. Currently, the genus comprises 22 recognized species, although only a portion of them are known to be clinically relevant in humans [13]. The *Enterobacter* genus is a member of the ESKAPE group of pathogens—an acronym encompassing *Enterococcus faecium*, *Staphylococcus aureus*, *Klebsiella pneumoniae*, *Acinetobacter baumannii*, *Pseudomonas aeruginosa*, and *Enterobacter* spp.—which are recognized as major contributors to healthcare-associated infections [14]. *Enterobacter* species, alongside other ESKAPE pathogens, are responsible for a broad spectrum of nosocomial infections, particularly in immunocompromised individuals, intensive care unit patients, and—less frequently—in community-acquired infections. Clinical manifestations may include bloodstream infections, urinary tract infections, respiratory tract infections, soft tissue infections, osteomyelitis, and infectious endocarditis [15]. Given this spectrum of organisms, in addition to providing effective clinical management, a key objective was to investigate the potential route of infection and understand how *Enterobacter* had colonized the paranasal sinuses of a healthy child.

Acute sinusitis caused by *Enterobacter* spp. has seen occasional emergence as a significant concern in pediatric populations, and more research papers have focused on the analysis of the transmission mechanisms, risk factors, and epidemiological trends associated with this type of infection in healthy children. Community-associated *Enterobacter* infections are increasingly recognized, particularly in urban areas. Key factors include environmental contamination, where the bacteria can persist in community environments, such as households or public spaces, facilitating transmission [16,17] and travel to regions with high rates of antibiotic resistance [17,18]. *Enterobacter* can spread through direct or indirect contact, particularly in settings with a high population density, such as schools or daycare centers. This mode of transmission is exacerbated by poor hygiene practices [19].

Although *Enterobacter* spp. is rarely reported as a causative agent of acute sinusitis, especially in immunocompetent individuals and children, their involvement has been documented in nosocomial cases, particularly in patients undergoing prolonged mechanical ventilation [20]. Several risk factors contribute to the development and spread of *Enterobacter* infections, such as inadequate hand hygiene of the staff, prematurity, mechanical ventilation, prolonged hospitalization, contaminated infant formula, parenteral nutrition, and the use of broad-spectrum antibiotics [21].

The therapeutic management of *Enterobacteriaceae*, including *Enterobacter* spp., remains challenging due to their evolving resistance profiles and reduced susceptibility to standard antimicrobials [22]. *Enterobacter* species often exhibit intrinsic resistance to ampicillin and many broad-spectrum cephalosporins, primarily due to the production of chromosomally encoded *AmpC* β-lactamases. Moreover, their ability to acquire mobile genetic elements—such as plasmids and transposons—has contributed to the emergence and dissemination of resistance to multiple antibiotic classes, including third-generation cephalosporins and carbapenems. This multidrug-resistant profile significantly complicates the selection of effective empirical and targeted antimicrobial therapies, particularly in vulnerable pediatric populations [23]. *Enterobacteriaceae* develop resistance to carbapenems primarily through three mechanisms: the production of hydrolytic enzymes, the overexpression of efflux pumps, and porin mutations [24]. In most *Enterobacter* species, β-lactamase production represents the primary mechanism responsible for resistance to β-lactam antibiotics. Notably, *Enterobacter aerogenes* and *Enterobacter cloacae* possess a high capacity to regulate and adapt these resistance mechanisms. Of particular concern is their ability to produce low levels of chromosomally encoded *AmpC* β-lactamase (Table 1), which confers resistance to first-generation cephalosporins [24].

Although *Enterobacter* species are rare causative agents of acute bacterial sinusitis, their presence has been documented more frequently in chronic rhinosinusitis, nosocomial settings, or following repeated antibiotic exposure, particularly in adults. In pediatric populations, the isolation of *Enterobacter* spp. remains exceptional and is typically associated with either underlying immunocompromise, hospital-acquired infections, or prior broad-spectrum antimicrobial use [1,3]. A few studies have reported *Enterobacteriaceae*, including *Enterobacter cloacae* and *Enterobacter aerogenes*, as part of mixed infections in both acute and chronic sinusitis cases, but the majority involved adults or patients with surgical histories such as sinus surgery or dental interventions [5]. In contrast, children with acute sinusitis typically harbor *Streptococcus pneumoniae*, *Haemophilus influenzae*, or *Moraxella catarrhalis*, and the isolation of *Enterobacter* may raise suspicion for antimicrobial resistance, biofilm formation, or secondary colonization in treatment-refractory cases [24,25]. The pediatric case reported here—an immunocompetent child with community-acquired sinusitis caused by *Enterobacter* spp.—therefore represents an unusual clinical finding that highlights the evolving microbiological landscape and the need for culture-guided therapy even in typical-appearing sinusitis presentations. While *Enterobacter* is not a primary pathogen in acute sinusitis, its role in chronic and complicated cases highlights the need for vigilance in monitoring emerging pathogens and resistance patterns [25].

Fortunately, in our case, despite the inherent concerns about resistance often associated with *Enterobacter*, the specific strain isolated from our patient proved susceptible to both ceftriaxone (the initial empiric therapy) and gentamicin, according to the Vitek2 analysis. Although genetic analysis detected potential inducible resistance markers like *ctxM15*, the phenotypic susceptibility validated the combined antibiotic regimen used post-surgery and likely contributed significantly to the patient’s successful recovery after the infection source was surgically addressed. The genetic analysis of bacteria detected from isolated cases, such as our pediatric case, is very important as key resistance genes may trace bacterial origins to their roots and predict medical outcomes and even complication rates. Carbapenemase genes like *blaNDM*, *blaIMP*, and *blaKPC* are the most common carbapenemase genes identified in resistant *Enterobacter* isolated from children [26] and are often responsible for empiric antimicrobial therapy failure and subsequent complications.

The severity of bacterial sinusitis significantly influences the decision to perform endoscopic sinus surgery (ESS) in pediatric patients. ESS is generally considered when medical management fails, particularly in cases of chronic rhinosinusitis (CRS) or when complications arise from acute sinusitis. The decision to proceed with surgery is complex and involves evaluating the severity of the condition, the presence of complications, and the impact on the child’s quality of life. The severity of sinusitis, particularly when it leads to complications, such as orbital or intracranial extension, often prompts the need for surgical intervention.

Indications for Surgery:•In CRS: ESS is often reserved for children with CRS who do not respond to maximal medical therapy, including antibiotics and nasal steroids. The failure of medical management is a primary indication for considering surgery [25,27].•Complications of acute sinusitis: Severe cases of acute sinusitis that lead to complications, such as orbital abscesses or intracranial involvement, often require surgical intervention. These complications are critical indicators for ESS, as they pose significant health risks [28,29].•Recurrent acute rhinosinusitis: In cases where children experience recurrent episodes of acute sinusitis despite medical treatment, surgery may be considered to prevent further complications and improve quality of life [29].

The presence of severe symptoms or complications, including orbital cellulitis and intracranial abscesses, often necessitates surgical intervention. These conditions are considered emergencies and require prompt surgical management [28,30]. Although surgical intervention is reserved for cases with orbital complications, such as large subperiosteal abscesses or orbital cellulitis unresponsive to medical therapy, endoscopic sinus surgery, including sinus drainage, may be necessary to eradicate or prevent further complications [31].

The decision to perform ESS must also take into consideration the child’s age and anatomical development. Younger children may have different surgical outcomes compared to older children, and the potential impact on facial growth is a consideration, although recent studies suggest minimal long-term effects [32]. ESS has a high success rate, often reported between 71% and 100%, with a low complication rate. This makes it a viable option for children who do not respond to other treatments [32].

Finally, the potential role of bacterial biofilms in *Enterobacter*-related sinusitis should be considered, especially in complicated or treatment-resistant cases. Biofilms are structured communities of bacteria encased in a self-produced extracellular matrix, allowing pathogens to persist on mucosal surfaces and evade host defenses and antibiotic therapy. *Enterobacter* spp., like other members of the *Enterobacteriaceae* family, are capable of forming robust biofilms, particularly in the presence of inflamed or damaged mucosa [33]. This ability has been demonstrated to occur in both in vitro and in vivo studies and may account for delayed or incomplete clinical responses, even in the absence of overt chronic sinusitis [34]. In pediatric cases—where biofilm-associated pathogens are less expected—the identification of *Enterobacter* should prompt clinicians to consider biofilm-mediated resistance mechanisms [35], particularly if genetic markers for inducible resistance are detected. The chronic aspect of inflammation observed during routine or follow-up endoscopy and the need for surgical debridement further support the possibility of biofilm involvement in such a case [36]. If biofilm-forming bacteria are suspected to be the culprit of an upper respiratory tract infection, emerging evidence supports the use of intranasal probiotic therapy, such as *Streptococcus salivarius*, for restoring microbial balance and enhancing mucosal immunity following the use of antibiotics or surgical interventions [37]. This approach may reduce the risk of recurrent infections and support mucosal healing by inducing competition between pathogenic species for adhesion sites [38]. In our case, a *Streptococcus salivarius* nasal spray was recommended post-discharge as a preventative strategy as we could not reference other similar literature findings or follow guidelines for this type of pediatric pathology.

## 4. Conclusions

Our case brings forth the importance of maintaining clinical awareness for atypical pathogens in acute atypical rhinosinusitis presentations, particularly when patients fail to respond to first-line therapies and in all cases that concern children especially. The identification of *Enterobacter* spp. in an otherwise healthy child without prior hospitalizations or chronic illness raises questions regarding community transmission pathways. Although traditionally associated with nosocomial settings, the presence of inducible resistance genes in this isolate suggests the potential emergence of new, community-acquired strains or alternative modes of transmission—such as poor hand hygiene combined with environmental exposure. Early surgical intervention (like endoscopic sinus surgery) to drain the infection, combined with appropriately chosen, culture-guided antibiotic therapy, is crucial for achieving a favorable and quick outcome in complicated or atypical cases. The possibility of biofilm formation by pathogens like *Enterobacter* should be considered in treatment-resistant or complicated cases.

This case reinforces the need for careful monitoring and the adaption of treatment plans based on clinical response and diagnostic findings, even in what might initially appear as routine pediatric sinusitis.

## Figures and Tables

**Figure 1 reports-08-00066-f001:**
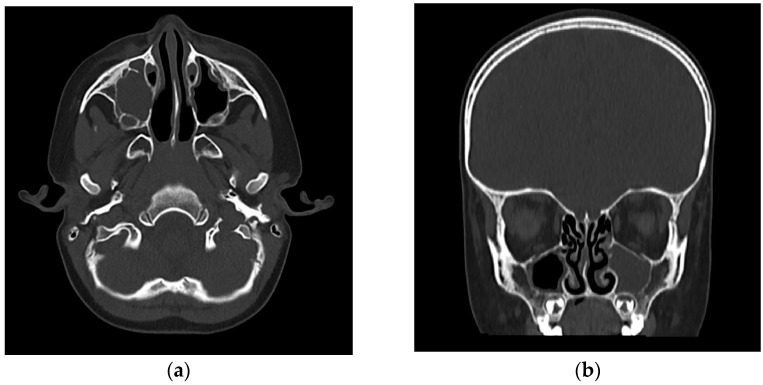
(**a**) Axial view CT (bone window): complete opacification of the right maxillary sinus, showing no evidence of orbital or bony involvement. Subtle subcutaneous soft tissue thickening and increased density were noted over the right infraorbital region, suggestive of mild localized edema. (**b**) Coronal view CT (bone window): complete opacification of the right maxillary sinus. The right anterior ethmoid air cells showed mild mucosal thickening. The lamina papyracea and orbital floor were intact, with no radiological evidence of intraorbital extension or abscess formation. The coronal section confirms the blockage of the right osteomeatal complex.

**Figure 2 reports-08-00066-f002:**
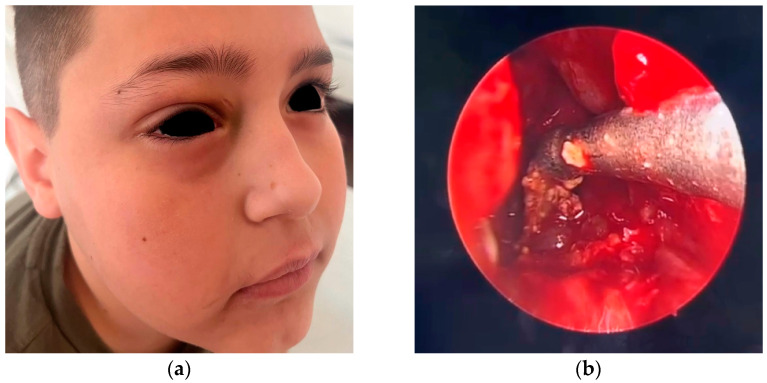
(**a**) Preoperative aspect demonstrates the persistent edema in the right palpebral region, particularly in the upper eyelid, with subtle darkening of the skin in the medial canthal region, suggesting early venous stasis or subcutaneous congestion; (**b**) endoscopic intraoperative aspect after performing right maxillectomy and suctioning of dense secretions from the right maxillary sinus.

**Table 1 reports-08-00066-t001:** Resistance mechanisms and associated genes within the *Enterobacter* genus.

Resistance Mechanism	Description	Genes/
AmpC Beta-Lactamases	Naturally produced by Enterobacter spp.; hydrolyze most β-lactams; expression can be inducible or constitutively high.	AmpC
Extended-Spectrum Beta-Lactamases	Acquired via plasmids; confer resistance to third-generation cephalosporins.	bla_CTX-M, bla_SHV, bla_TEM
Carbapenemases	Carbapenem-resistant strains appear due to plasmid-mediated genes acquired.	bla_KPC, bla_NDM, bla_VIM, bla_IMP, bla_OXA-48
Efflux Pumps and Porin Loss	Increased efflux (e.g., AcrAB-TolC) and decreased porin channels (e.g., OmpC, OmpF) reduce antibiotic uptake and increase resistance.	AcrAB-TolC, OmpC, OmpF
Other Resistance Genes	Resistance to specific drug classes (aminoglycosides, fluoroquinolones, and colistin).	aac(6′)-Ib, qnr, mcr

## Data Availability

The original contributions presented in this study are included in the article. Further inquiries can be directed to the corresponding author.

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
