# Peer review of "Unilateral Palpebral Edema as a Central Sign of Acute Enterobacter-Associated Rhinosinusitis in a 5-Year-Old: A Rare Pediatric Case"

_reports, 2025, doi:10.3390/reports8020066_

Round 1
Reviewer 1 Report
Comments and Suggestions for Authors
The case report is interesting because it describes a rare case of maxillary sinusitis from Enterobacter cloacae in a 5-year-old immunocompetent child. The authors have made a complete and thorough description of the case and the literature.
However, I have some questions:
what were the symptoms before the appearance of the eyelid edema?
How did the child describe the symptom that you called didplopia?
Did the surgeons use a nasal packing to contain the post-surgical bleeding?
Good job.
Author Response
Reviewer Comment 1: What were the symptoms before the appearance of the eyelid edema?
Response:
We thank the reviewer for this insightful question. We have modified the manuscript so that it details more on this.
Reviewer Comment 2: How did the child describe the symptom that you called diplopia?
Response:
Thank you for highlighting this. We have added clarifying notes describing the child's double vision problems.
Reviewer Comment 3: Did the surgeons use a nasal packing to contain the post-surgical bleeding?
Response:
We appreciate this important technical question. As now indicated in the revised manuscript, we did use nasal packing to maintain meatus patency.
Reviewer Comment 4: Good job.
Response:
We sincerely thank the reviewer for this kind remark and for the thoughtful suggestions, which have helped improve the clarity and completeness of our manuscript.
Reviewer 2 Report
Comments and Suggestions for Authors
The authors have presented an interesting case report of acute rhino sinusitis of an unusual organism in a healthy host. They have presented a detailed account of the case and the treatment. Minor changes are suggested to improve the case report
Lines 59-60: Please modify the sentence 'empiric therapy seldom succeeds.." and not seldom fails. If it seldom fails, then there is no need for further investigation
Please give aa reference on the safety and efficacy of the use of probiotic nasal spray. Please mention details on how long it was used and how frequently. What was the dose used. Were any nasal saline sprays/wash was advised post surgery
Please discuss and reference other case reports or studies with similar organisms causing acute rhinosinusitis in a normal healthy host.
Author Response
Reviewer Comment:
Lines 59–60: Please modify the sentence “empiric therapy seldom succeeds...” and not “seldom fails.” If it seldom fails, then there is no need for further investigation.
Response:
Thank you for this observation. We have corrected the sentence in the revised manuscript accordingly.
Reviewer Comment:
Please give a reference on the safety and efficacy of the use of probiotic nasal spray. Please mention details on how long it was used and how frequently. What was the dose used. Were any nasal saline sprays/wash advised post-surgery.
Response:
Thank you for this valuable suggestion. We have added references supporting this sort of approach. Saline was prescribed, for at-home use, as the text mentions but we also added the regimen or probiotic spray recommended in text.
Reviewer Comment:
Please discuss and reference other case reports or studies with similar organisms causing acute rhinosinusitis in a normal healthy host.
Response:
We appreciate this important point. We have some discussions as to why that might happen in an otherwise healthy child but as to this point, were not able to uncover any literature reference detailing a similar experience - for this matter, we felt that this type of disease presentation should be reported.
Thank you for taking your valuable time and reviewing our article.
Reviewer 3 Report
Comments and Suggestions for Authors
Dear Authors,
I read the article with interest and think the it's good structure.
Indeed, pediatric acute rhinosinusitis is caused in most cases by Streptococcus pneumoniae, Haemophilus influenzae and Moraxella tarrhalis. The involvement of Enterobacter is rare and it may be worth reporting your clinical case to increase awareness in the ENT community of this clinical entity.
I appreciate the discussion session where you summarized the indications for surgery in pediatric acute rhinosinusitis.
I think your article can be accepted in this form without changes.
Author Response
Thank you very much for reading the article and taking your time to make such appreciative comments.